# Alterations in Calcium Handling Are a Common Feature in an Arrhythmogenic Cardiomyopathy Cell Model Triggered by Desmosome Genes Loss

**DOI:** 10.3390/ijms24032109

**Published:** 2023-01-20

**Authors:** Marta Vallverdú-Prats, David Carreras, Guillermo J. Pérez, Oscar Campuzano, Ramon Brugada, Mireia Alcalde

**Affiliations:** 1Cardiovascular Genetics Center, Biomedical Research Institute of Girona, 17190 Salt, Spain; 2Department of Medical Sciences, Universitat de Girona, 17003 Girona, Spain; 3Centro de Investigación Biomédica en Red de Enfermedades Cardiovasculares (CIBERCV), 21005 Madrid, Spain; 4Hospital Josep Trueta, 17007 Girona, Spain

**Keywords:** Arrhythmogenic Cardiomyopathy, calcium handling, CRISPR/Cas9, desmosomes, HL1

## Abstract

Arrhythmogenic cardiomyopathy (ACM) is an inherited cardiac disease characterized by fibrofatty replacement of the myocardium. Deleterious variants in desmosomal genes are the main cause of ACM and lead to common and gene-specific molecular alterations, which are not yet fully understood. This article presents the first systematic in vitro study describing gene and protein expression alterations in desmosomes, electrical conduction-related genes, and genes involved in fibrosis and adipogenesis. Moreover, molecular and functional alterations in calcium handling were also characterized. This study was performed d with HL1 cells with homozygous knockouts of three of the most frequently mutated desmosomal genes in ACM: *PKP2, DSG2,* and *DSC2* (generated by CRISPR/Cas9). Moreover, knockout and N-truncated clones of DSP were also included. Our results showed functional alterations in calcium handling, a slower calcium re-uptake was observed in the absence of *PKP2, DSG2*, and *DSC2*, and the *DSP* knockout clone showed a more rapid re-uptake. We propose that the described functional alterations of the calcium handling genes may be explained by mRNA expression levels of *ANK2, CASQ2, ATP2A2, RYR2*, and *PLN*. In conclusion, the loss of desmosomal genes provokes alterations in calcium handling, potentially contributing to the development of arrhythmogenic events in ACM.

## 1. Introduction

Arrhythmogenic cardiomyopathy (ACM) is a rare heart disease characterized by the progressive loss of cardiomyocytes and its replacement by fibrofatty tissue. It affects the right ventricle, but biventricular and left-dominant forms predominantly have also been reported [1,2]. The principal symptoms are ventricular arrhythmias and sudden cardiac death, which are present in up to 50% of cases [3,4]. ACM affects one in 2000–5000 individuals, and it has been reported that deleterious variants of genes encoding desmosomal proteins are the main cause of ACM [5].

Desmosomes are essential for cardiomyocyte integrity because they provide cells with the ability to resist mechanical forces by joining cells together and connecting extracellular contacts with internal intermediate filaments [6]. These structures are located in intercalated discs, and they are composed of plakophilin2 (*PKP2*), desmoplakin (*DSP*), desmoglein2 (*DSG2*), desmocollin2 (*DSC2*), and plakoglobin (*JUP*). Pathogenic variants in any of these genes have been associated with ACM, with *PKP2* the most common mutated gene in ACM patients (10–45%), followed by *DSP* (10–15%), *DSG2* (7–10%), and *DSC2* (2%) [7].

A general molecular feature of the disease is alterations in expression levels of desmosomes or genes related to electrophysiologic functions of the heart. In that sense, it has been described that electrical coupling and intercellular adhesion in the heart are connected by a functional unit called connexome, which is constituted by voltage-gated sodium channel (Nav1.5), Connexin-43 (Cx43) and desmosomes [8,9]. It is known that there is an overlapping between Brugada Syndrome and ACM due to these interactions found in the connexome [10,11].

Moreover, fibrosis and adipogenesis play an important role in ACM, with transforming growth factor beta 1 (TGFB1) and peroxisome proliferator-activated receptor gamma (PPARγ), respectively, the key genes of these pathways [12]. However, part of the pathophysiological mechanism in ACM remains to be clarified.

To date, a great number of animal and cell models with genetic alterations in desmosomal genes have been generated [12,13,14]. The HL1 cell line is one of the most preferred in vitro models due to its capacity to maintain differentiated cardiac properties at morphological, biochemical, and electrophysiological levels after several passages [15]. HL1 was generated from the AT-1 atrial cardiomyocyte tumor lineage, and it was the first ACM cell model [14].

Because of all these available models, it has been possible to identify gene-specific molecular alterations in ACM. However, not all desmosomal genes are equally studied. There are many studies describing molecular and functional consequences of the loss of *PKP2*, both in vitro and in vivo, but the number of studies is reduced for *DSC2* or *DSG2* in ACM. It is still unknown if the alterations described in *PKP2*, such as disruption in calcium handling [16,17,18,19] or downregulation of *Cx43* expression [14,20,21], are shared by all desmosomal genes.

Our study aims to elucidate common molecular alterations triggered by the loss of desmosomal genes to unmask key factors that may be important for ACM development. Specifically, we present the first systematic study characterizing molecular alterations using a consistent in vitro model to study the main desmosomal genes. Specifically, we studied gene expression of desmosomal, calcium handling, electrical conduction, *TGFB1*, and *PPARγ* genes, and functional changes in calcium homeostasis with the loss of *PKP2*, *DSG2, DSC2,* and *DSP* using CRISPR-edited HL1 cell lines.

## 2. Results

### 2.1. Generation and Validation of Gene-Specific Knockout Clones PKP2, DSG2 and DSC2

The *PKP2*-KO, *DSG2*-KO, and *DSC2*-KO groups comprised four HL1 clones for each edited gene, *PKP2, DSG2*, and *DSC2*, respectively. All of the clones presented frameshifts leading to a premature termination codon (PTC) within the 5′ region of the three desmosomal genes studied. None of the edited clones produced detectable levels of the truncated protein (Appendix A). Genotypes of all 12 edited clones are shown in Appendix A.

### 2.2. Alterations in the Expression Levels of Desmosomal Genes

The expression levels of mRNA and protein of the five desmosomal genes in the *PKP2*-KO, *DSC2*-KO, and *DSG2*-KO groups were evaluated by RT-PCR and WB. At the mRNA level, all 12 edited clones presented undetectable expression levels of *DSC2* (Figure 1A), with *DSC2* downregulation as the only common feature. Additionally, gene-specific alterations were also found, depending on the knocked-out gene, with *DSC2*-KO the only group with no downregulation in the desmosomal genes. The relative quantifications (RQs) of the RT-PCR are shown in Appendix A.

Consistently, at the protein level, DSC2 downregulation was also the only common alteration. However, other alterations were revealed. *PKP2*-KO clones showed undetectable levels of DSC2 and canonical DSG2 (130 KDa) and downregulated DSP and PG, with decreased levels of all desmosomal proteins (Figure 1B). Moreover, *DSG2*-KO clones showed decreased levels of all proteins except for DSP (Figure 1C). Finally, the only alteration in the desmosomal expression found in *DSC2*-KO clones was the downregulation of the DSP protein (Figure 1D). Appendix A shows the *p*-value of the statistical analysis of mRNA and protein expression levels of all edited clones.

### 2.3. Alterations in the Calcium Cycle

#### 2.3.1. Expression Levels

To detect common alterations regarding calcium handling in the edited clones, both mRNA and protein expression levels of calcium genes known to be decreased in the absence of *PKP2* [16] were studied.

At the mRNA level, all *PKP2, DSG2*, and *DSC2* edited clones shared downregulation of *ANK2, ATP2A2*, and *CASQ2* (Figure 2A). Gene-specific alterations were also found, with *PKP2*-KO the group with the greatest decrease in the calcium handling genes, showing significant downregulation in all the studied genes except for *PLN*. In contrast, *DSC2*-KO clones, apart from the shared alterations, only showed downregulation of *TRDN* and were the group with the least molecular alterations. Interestingly, although *PLN* did not show significant differences in any of the desmosomal KO groups, it exhibited a tendency of upregulation in all of them. RQs of the RT-PCR results are shown in Appendix A.

At the protein level, ANK2 and RYR2 were found to be significantly downregulated in the three groups, which was a common feature among all the clones (Figure 2B). Replicates of the RYR2 WB are shown in the Appendix A. Appendix A shows the *p*-value of the statistical analysis of the mRNA and protein expression levels of all edited clones.

#### 2.3.2. Calcium Imaging

Edited clones were evaluated functionally for calcium handling homeostasis. First, the characteristics of calcium transients elicited by a local pulse of caffeine (10 mM caffeine, 1 s duration) were studied. Second, the rate of the cytosolic calcium rise and the kinetics of the decay were evaluated. To this end, traces of fluorescence of a colorant over time representing cytosolic calcium transients were measured. The number of traces obtained per cell line is shown in Appendix A, and the final plot representing the averaged traces of all KO groups is presented in Figure 3A.

All three KO groups presented similar kinetics but were significantly different compared to WT clones. Additionally, hierarchical statistical analysis was performed on six parameters of the calcium transient: half-width duration, rise time 10–50%, decay time 50–10%, rise time 10–90%, decay time 90–10%, and decay time 90–50% (Figure 3B, Appendix A). Interestingly, all the clones presented a significantly longer half-width duration compared to WT clones, indicating a slower calcium removal. Furthermore, *DSG2*-KO and *DSC2*-KO also presented a significantly longer rise time of 10–90% and decay time of 90–50%, and only *DSG2*-KO showed a significantly longer decay time of 90–10% compared with WT clones. These increased times indicate both slower release and re-uptake in the KO clones *DSG2* and *DSC2*. All values of calcium transients are shown in Appendix A.

### 2.4. Alterations in Electrical Conduction Related Genes Cx43 and Nav1.5

Cx43 and Nav1.5 (encoded by SCN5A) expression levels were evaluated to detect possible shared alterations related to electrical functionality in the edited clones. At the mRNA level, *Cx43* was downregulated in all the clones studied (Figure 4A), although only *DSG2*-KO showed a significantly lower expression of Cx43 protein (Figure 4B). Replicates of the CX43 WB are shown in the Appendix A. *SCN5A* showed downregulation only in the *PKP2*-KO clones at the mRNA level (Figure 4A). However, all edited clones showed undetectable levels of Nav1.5 protein. RQs of the RT-PCR are shown in Appendix A. Appendix A shows the *p*-value of the statistical analysis of the mRNA and protein expression levels of all edited clones.

### 2.5. Fibrosis and Adipogenesis: Alterations in TGFB1 and PPARγ Genes

The expression levels of *TGFB1* and *PPARγ* genes, as key factors of fibrosis and adipogenesis, were evaluated by RT-PCR. The results revealed that *TGFB1* was significantly upregulated in *DSG2*-KO and *DSC2*-KO groups, while *PKP2*-KO showed only a tendency in the same direction (Figure 5). Appendix A shows the *p*-value of the statistical analysis, and the RQs of the RT-PCR are shown in Appendix A. Additionally, higher levels of *PPARγ* were detected in *PKP2*-KO, *DSG2*-KO, and *DSC2*-KO groups (Appendix A). RQs could not be calculated for *PPARγ* as no expression was detected in any of the four WT clones (cycle threshold (CT) >35 cycles, CTs mean= 36.18), while edited clones showed CT <35 cycles (*PKP2*-KO CT mean = 32.5, *DSG2*-KO CT mean = 31.06, and *DSC2*-KO CT mean = 33.65).

### 2.6. Molecular and Functional Evaluation of N-Truncated DSP and DSP-KO

The *DSP*-KO edited clones, in contrast to the *PKP2*-KO, *DSC2*-KO, and *DSG2*-KO clones, triggered the re-initiation of translation and were able to synthesize N-DSP [22]. Molecular and functional characterization of N-*DSP* and *DSP*-KO in this group was performed qualitatively as statistical analyses were not possible because one *DSP*-KO clone did not trigger re-initiation of translation. In contrast, the remaining clones synthesized N-*DSP* [22].

#### 2.6.1. Expression Levels

mRNA levels of N-*DSP* and *DSP*-KO clones were analyzed by RT-PCR for CACNA1C and PLN, which were not included in our previous study [22]. This suggested PLN downregulation in *DSP*-KO (Figure 6A), in contrast to the higher *PLN* levels observed in *PKP2*-KO, *DSG2*-KO, and *DSC2*-KO clones (Figure 2A). RQs are shown in Appendix A.

WB was performed for N-*DSP* and *DSP*-KO clones to analyze DSC2, ANK2, and Nav1.5 protein levels, which were undetectable for the *PKP2*-KO*, DSG2*-KO, and *DSC2*-KO clones and thus a possible common feature of desmosomal genes loss. From these proteins, *DSP*-KO clones had undetectable DSC2 levels, while N-*DSP* showed normal DSC2 expression (Figure 6B). However, loss of ANK2 and Nav1.5 was not observed for *DSP*-KO or N-*DSP* clones (Figure 6B). Finally, DSG2 protein levels were also studied, but none of the clones showed any change in the levels of total DSG2 (Figure 6B).

#### 2.6.2. Calcium Imaging

Finally, a 10-mM caffeine peak was performed on N-*DSP* and *DSP*-KO clones to describe their calcium-handling homeostasis at a functional level. The number of traces per clone is shown in Appendix A, and the peak is shown in Figure 7A. Both clones, N*-DSP* and *DSP*-KO, presented different kinetics compared with the other three groups evaluated (*PKP2*-KO, *DSG2*-KO, and *DSC2*-KO) (Figure 3A). The N-*DSP* peak was similar compared with the WT clones, showing an analogous amplitude. However, *DSP*-KO presented slightly different kinetics compared to both WT and N-*DSP* clones with a shorter amplitude, indicating a quicker calcium removal. Results for the six kinetic parameters are represented in Figure 7B and Appendix A. All clones had similar values for all parameters with a high standard deviation. However, the half-width may have been shorter in *DSP*-KO compared with WT and N-*DSP* clones (Figure 7B, Appendix A). Hierarchical statistical analysis was not performed due to the limited number of clones.

## 3. Discussion

ACM is an inherited cardiomyopathy mainly caused by rare pathogenic variants in genes encoding desmosomal proteins. Although there are several studies describing molecular alterations found in ACM cells and animal models, most of them are only focused on PKP2, the main gene currently associated with ACM. Few studies, including all desmosomal genes using the same technical approach, have been published so far. For this reason, the present study aimed to elucidate ACM common alterations shared by the loss of desmosomal genes. Here, we presented the first systematic in vitro study using CRISPR-edited HL1 cells to characterize gene expression profiles and functional alterations following the loss of *PKP2, DSG2, DSC2*, and *DSP*.

### 3.1. DSC2 Downregulation

This study revealed a dramatic downregulation of *DSC2* as a common feature of the ACM cellular models, shared by all desmosomal KOs studied. These results suggested that *DSC2* downregulation may be a common molecular feature of the ACM cellular phenotype. In this sense, previous studies in explanted hearts of ACM indicated the same findings, showing a general decrease of either *DSC2* mRNA levels [23] or DSC2 protein levels [24]. However, the molecular mechanisms leading to this common *DSC2* downregulation remain unknown, and further studies will be needed.

Interestingly, some previous studies using ACM heart samples found divergent results in desmosomal protein expression. One study described the downregulation of all desmosomal genes [25], while another only found differences in some of them [23], and a third did not find significant differences at all [24]. These discrepancies might be explained by the different causal genes of the ACM samples. Our systematic study showed downregulation of *DSC2* was a common feature caused by the absence of *PKP2, DSG2, DSC2*, or *DSP*. However, it is still unclear if this alteration was also shared by different variants other than those with PTCs or by other ACM-associated genes. Moreover, here, no other common alterations in desmosomal expression genes have been described, suggesting that the downregulation of the other desmosomal genes found in the present study might be gene-specific.

### 3.2. Cx43 and Nav1.5 Downregulation

To date, alterations in Cx43 and Nav1.5 levels or sodium currents have been widely associated with *PKP2* [14,20,21,26,27,28], or *DSP* [29,30,31,32] decreases or silencing, while they are quite unknown for *DSG2* and *DSC2*. It has been reported that a null allele in *DSG2* caused regional differences regarding Cx43 expression but no significant differences in the total Cx43 protein content [33]. Moreover, another publication indicated that a null mutation in *DSC2* was associated with gap junction remodeling and decreased Cx43 in the intercalated discs [34]. However, only one patient was involved in that study, and possible alterations in Nav1.5 levels remain unknown for *DSC2* and *DSG2* mutations. Our study showed a significant downregulation of Cx43 at the transcriptional level among all *PKP2*-KO, *DSG2*-KO, and *DSC2*-KO clones. This downregulation was also validated at the protein level in *DSG2*-KO clones. At the same time, no significant differences were found for *PKP2*-KO and *DSC2*-KO clones, although our results suggested a tendency to downregulation of protein levels.

Nav1.5 protein was not detected in any of the *PKP2*-KO, *DSG2*-KO, or *DSC2*-KO groups, indicating that, as previously reported for *PKP2* loss [26,28], the absence of *DSG2* or *DSC2* may also produce alterations in the sodium current. Additionally, unlike the other groups, the *DSP*-KO clones presented detectable levels of Nav1.5. However, our results did not determine if the total levels of Nav1.5 were significantly different from WT, as described earlier [32], because of the very limited number of DSP-KO clones.

### 3.3. TGFB1 and PPARγ Upregulation

Our results showed an increase in *TGFB1* and *PPARγ* shared by the *PKP2*-KO, *DSG2*-KO, and *DSC2*-KO clones. Specifically, the results demonstrated that TGFB1 was significantly increased in *DSG2*-KO and *DSC2*-KO clones, suggesting that the loss of *DSG2* and *DSC2* could activate fibrosis. It was previously reported that the loss of *PKP2* increased the expression of *TGFB1* [35]. Interestingly, Dubash et al. demonstrated that it was the loss of *DSP* expression in a *PKP2* knockdown that provoked the increased expression of TGFB1 since the rescue of *DSP* expression restored normal levels of *TGFB1* in that model [35]. Our results of the *DSC2*-KO clones showed that increased levels of *TGFB1* and a reduction of *DSP* levels were in concordance with the proposed mechanism in this previous publication. However, our results of the *DSG2*-KO clones also showed increased levels of *TGFB1* but unaltered levels of *DSP*, suggesting that the *TGFB1* upregulation may also occur through a *DSP*-loss-independent manner. More studies would be needed to elucidate the underlying mechanisms in these clones.

An increase in *PPARγ* expression was also shared by the *PKP2*-KO, *DSG2*-KO, and *DSC2*-KO groups. These results point in the same direction as previously published studies. It has been reported that the right ventricles of ACM patients expressed higher levels of *PPARγ,* independently of the mutated gene [36]. Furthermore, cell models with alterations in *PKP2* or *DSC2* also showed increased levels of *PPARγ* [25,37,38]. Thus, our results support the idea that increased *PPARγ* is a common molecular feature in the ACM cellular phenotype. It is triggered by either a *PKP2, DSG2*, or *DSC2* loss, potentially leading to an adipogenic process.

### 3.4. Downregulation in Calcium Handling Gene Expression

In the last few years, a small number of studies have been published describing calcium handling gene alterations associated with a desmosomal protein loss [16,18,19,39]. Three of them were focused on the loss of *PKP2*, which is the most studied gene in this field. It is well described that transcription of genes involved in the calcium cycle requires PKP2 [16]. The present study characterized, for the first time, the permanent loss of different desmosomal proteins using CRISPR editing in the same in vitro model, thus avoiding differences between the selected models.

The main shared feature among *PKP2*-KO, *DSG2*-KO, and *DSC2*-KO clones was the dramatic downregulation of RYR2 and ANK2 proteins. Our results supported previous studies showing a significant decrease in RYR2 and ANK2 in a *PKP2*-KO mouse model [16,19] and also revealed these downregulations to be a common feature between *PKP2* loss and cadherin (*DSG2* and *DSC2*) loss.

Finally, the *DSP*-KO clones exhibited detectable levels of ANK2 protein, while the *PKP2*-KO, *DSG2*-KO, and *DSC2*-KO groups did not. Moreover, the shared tendency for the upregulation of *PLN* among the *PKP2*-KO, *DSG2*-KO, and *DSC2*-KO groups was remarkable, even more so given that *DSP*-KO showed the opposite tendency. These data together suggest that *DSP* loss might have different impacts on calcium handling gene expression.

### 3.5. Slower Re-Uptake of Calcium with PKP2, DSG2, or DSC2 Loss

Several studies have been published describing calcium cycling alterations using different in vivo and in vitro experimental approaches [16,18,39]. Our study explored these functional alterations triggered by desmosomal protein loss using, for the first time, the same in vitro system for all the main genes. Our results with *PKP2*-KO, *DSG2*-KO, and *DSC2*-KO clones showed similar calcium transient kinetics with a slower re-uptake of calcium with a significantly longer duration, while the *DSP*-KO clones presented completely different kinetics with a shorter amplitude. Our results are in concordance with previously published studies showing that the loss of *PKP2* caused a delayed amplitude of the peak in a mouse model [16]. However, regarding the loss of *DSG2,* it was previously shown to produce a shorter amplitude of the peak in heterozygous cardiomyocytes derived from iPSC cells [39]. These discordant results may be explained by the heterozygosity of the previously reported model and the different methodology used to measure calcium transients.

Taking together the massive dysregulation of calcium handling gene expression and subsequently altered calcium transient kinetics, we hypothesized putative molecular mechanisms underlying these results. *PKP2*-KO, *DSC2*-KO, and *DSG2*-KO clones presented shared downregulated levels of *ATP2A2, CASQ2, RYR2*, and *ANK2*. *ATP2A2* encodes SERCA2, and a decrease of this protein could be responsible for the delay in the re-uptake of calcium to the sarcoplasmic reticulum. *CASQ2* encodes calsequestrin-2, which is located inside the sarcoplasmic reticulum and acts as a calcium buffer, regulating the calcium release by *RYR2* [40]. Thus, the downregulation of *CASQ2* and *RYR2* might be the mechanism underlying the dysregulation of calcium release, shown as a significantly increased rise time in *DSG2*-KO and *DSC2*-KO clones. Interestingly, although *CASQ2* and RYR2 were also downregulated in *PKP2*-KO, the rise time was not significantly increased in this group. For this reason, the *PKP2* loss might trigger other additional mechanisms associated with these functional alterations. In this sense, it was previously published that the loss of *PKP2* triggered the phosphorylation of the RYR2 T2809 residue that led to an increased sarcoplasmic reticulum calcium load and higher diastolic calcium concentration due to a function gain of RYR2 [18]. This gain of function of RYR2 could compensate for the decreased levels of RYR2 protein in *PKP2*-KO clones, and this may explain the unaltered rise time. Therefore, our results are compatible with this previous idea that PKP2 loss may trigger specific mechanisms related to RYR2 phosphorylation that was not found in the other desmosomal gene clones.

ANK2 was commonly downregulated in *PKP2*-KO, *DSC2*-KO, and *DSG2*-KO clones. This protein plays an essential role in the localization and membrane stabilization of NCX1 (encoded by *SLC8A1*) and SERCA2 [40]. Therefore, we hypothesize that decreased ANK2 might be involved in the delayed re-uptake of calcium because NCX1 and SERCA2 may be less efficient in returning calcium to the extracellular medium and to the sarcoplasmic reticulum, respectively. In agreement with this hypothesis, DSP-KO clones, which showed detectable levels of ANK2 protein, presented a slightly shorter amplitude in the calcium peak. Therefore, ANK2 could be causing these differences in peak amplitude in the studied clones. In fact, previous studies showed that heterozygous ANK2-KO mice presented more frequent Ca2+ sparks and waves compared with WT mice [41,42]. Ca2+ waves are caused by a higher Ca2+ content in the sarcoplasmic reticulum [43], indicating that the absence of ANK2 causes dysregulation of calcium homeostasis.

Additionally, PLN is an inhibitor of the activity of ATP2A2, decreasing SERCA affinity for calcium in its unphosphorylated state [44]. An increased expression of PLN may reduce ATP2A2 activity, producing a delay in the re-uptake of calcium into the sarcoplasmic reticulum and thus causing the delayed amplitude of the peak in *PKP2*-KO, *DSG2*-KO, and *DSC2*-KO clones. However, in the *DSP*-KO clones, downregulated levels of PLN may trigger the higher activity of ATP2A2, leading to a more rapid re-uptake of calcium to the sarcoplasmic reticulum. Therefore, changes in *PLN* mRNA expression levels could explain the functional alterations in all clones.

Moreover, it is important to take into account that *PLN* is an ACM-associated gene classified as definite by ClinGen [45]. Specifically, there are several studies of the variant *PLN* R14del, which is prevalent in the Netherlands. It presents a stronger affinity for SERCA2, suggesting that the inhibition of SERCA2 is higher in those patients. This equates to a risk for malignant ventricular arrhythmias [46,47,48]. Regarding its role in calcium handling, it has been shown that it causes a slower SR Ca+2 re-uptake [49]. Taking this into account, the calcium handling dysregulation underlying ACM caused by *PLN* R14del or by the absence of *PKP2, DSG2*, or *DSC2* might be similar. More studies are needed in that direction to elucidate if those ACM-causal genes share molecular and functional alterations in calcium handling.

Regarding the relation between ACM clinical features and these molecular and functional detected alterations in calcium handling, it is interesting to add some related data. On the one hand, it has been observed an association between decreased SERCA2 and heart failure [50]. In that sense, it has been developed gene therapy for heart failure based on the overexpression of SERCA2. There is data that confirm its effectiveness [51,52], but, for the moment, it has not been validated in a large-scale clinical trial [53,54]. In the present study, *PKP2, DSG2,* and *DSC2*-KOs experiment with a decrease in ATPase expression (codified for SERCA), indicating that the absence of those genes may be linked to heart failure, a clinical feature present in ACM phenotype.

On the other hand, the molecular and functional detected alterations may also be related to dysregulation of the Ca^2+^ content, which is associated with different clinical alterations [43]. A high Ca^2+^ content in the SR could contribute to propagating a wave of Ca^2+^ induced—Ca^2+^ release increasing the propensity to arrhythmias, while a low Ca^2+^ content is associated with heart failure [43]. Our experimental approach was not able to determine the Ca^2+^ content of each cell line. Still, for future experiments, it would be interesting to measure it to see if there is any association between arrhythmias or heart failure with the loss of a determinate desmosomal protein via calcium handling.

Finally, our study also demonstrated that N-DSP proteins were produced by re-initiation of translation triggered by a 5′-PTC, with no major alterations in the expression profile [22], and were also revealed to be fully functional in terms of calcium handling.

## 4. Materials and Methods

### 4.1. sgRNA Design and Cloning into the Cas9 px458 Vector

The Benchling web tool (BiologySoftware, 2018; retrieved from https://benchling.com) was used to design the sgRNAs of the four desmosomal genes in the first 160 codons of the sequences for editing by CRISPR/Cas9. sgRNAs with high scores and a low off-target number were selected: PKP2 (5′-GTATGTCTACAAGCTACACG-3′, FW); DSP (5′-CCACCCGCGGATCAACACGC-3′, FW); DSG2 (5′-TGGCGCGGAGCCCGGGTGAC-3′, FW); DSC2 (5′-GCTGTGGGATCTATGCGCTCC-3′, FW). The px458 vector (plasmid #48138, Addgene, Teddington, UK), which encodes Cas9 wild-type (WT), was digested by BbsI-HF (R3539S, New England BioLabs, Ipswich, MA, USA) at 37 °C overnight and ligated by T4 DNA ligase (M0202L, New England BioLabs, Ipswich, MA, USA) for 1 h at room temperature with the sgRNA previously annealed (sense and antisense). Annealing of sgRNAs was performed with T4 PNK (M0201S, New England BioLabs, Ipswich, MA, USA) using the following thermocycler program: 30 min at 37 °C, 5 min at 95 °C, and 94 to 25 °C decreasing 1 °C per 12 s. DH5alpha competent cells (18265-017, Invitrogen, Waltham, MA, USA) were transformed with the sgRNA-px458 vector for 30 min on ice and 45 s at 42 °C. DNA was extracted using a Plasmid Midi Kit (12143, Qiagen, Hilden, Germany).

### 4.2. HL-1 Cell Culture and Electroporation

HL1 cells were cultured as described previously [15] at 37 °C under 5% CO2 on fibronectin-gelatin-coated slides in Claycomb medium (51800C, Sigma, St Louis, MO, USA) supplemented with 10% fetal bovine serum (10270106, GIBCO, Waltham, MA, USA), 100 U/mL penicillin, 10 mg/mL streptomycin (P4333-100 ML, Sigma, St Louis, MO, USA), 2 mM L-glutamine (35050061, Thermo, Waltham, MA, USA), 0.1 mM norepinephrine (A9512, Sigma, St Louis, MO, USA), and 0.3 mM ascorbic acid (A7631, Sigma, St Louis, MO, USA). Plasmids were nucleofected into HL-1 cells in suspension by using the Amaxa Cell Line Nucleofector Kit V (VCA-1003, Lonza, Basel, Switzerland); 106 cells per condition were transfected by adding 4 ug of the vector. Next, cells were seeded into 24-well plates, and after 48 h, they were diluted by seeding 10,000 cells on a P100 and 5000 cells on a 6-well plate. When colonies started growing, they were picked and seeded on a 24-well plate. Cells were expanded and frozen in a vial (with Claycomb medium and 10% dimethyl sulfoxide (D2650-5X5ML, Sigma, St Louis, MO, USA)) and a pellet to extract gDNA.

### 4.3. gDNA Extraction and Sanger Sequencing

To extract gDNA from the HL1 clones, QuickExtract (QE09050, Lucigen, Middleton, WI, USA) was used. For this process, 20 µL of the reagent was added to each pellet and vortexed for 13 s. Samples were incubated at 65 °C for 6 min, vortexed for 15 s, and incubated at 98°C for 2 min. Primers, PCR conditions, and kits used are listed in Appendix A. Next, ExoZap cleaning (7200100-1000, Ampliqon, Odense, Denmark) and BigDye reactions (4336911, Applied Biosystems, Waltham, MA, USA) were performed. DNA was precipitated by adding sodium acetate and 70% ethanol diluted in formamide. Samples were sequenced using a 3500 Genetic Analyzer (Applied Biosystems, Waltham, MA, USA). Sequencing Analysis Software 7 was used to analyze the sequences.

### 4.4. RNA Extraction and Real-Time PCR (RT-PCR)

Total RNA was purified using the RNeasy Mini Kit (74106, Qiagen, Hilden, Germany) according to the manufacturer’s instructions. The prior reverse transcription reaction was performed with an additional step of DNase I treatment and with gDNA Wipeout buffer. Reverse transcription reactions of RNA were performed using the QuantiTect Reverse Transcription Kit (205313, Qiagen, Hilden, Germany). A final volume of 20 µL of reverse transcription reaction was obtained by mixing 1 µg of total RNA with 4 µL of RT buffer, 1 µL of primer mix, and 1 µL of reverse transcriptase in nuclease-free water. cDNA was analyzed with real-time PCR reactions using a KAPA SYBER FAST Universal Kit (KK4602, KAPA Biosystems, St Louis, MO, USA) for desmosomal, calcium handling (except CACNA1C), Cx43, and SCN5A genes. For CACNA1C, TGFB1, and PPARγ, TaqMan Fast Advanced Master Mix (4444557, Applied Biosystems, Waltham, MA, USA) was used. RPLP0 was used as a housekeeping gene for both methods, and all data were analyzed using the QuantStudio™ Real-Time PCR System and Cloud Software (ThermoFisher, Waltham, MA, USA). The obtained results were analyzed statistically using SPSS by using the Mann–Whitney U test for comparisons between two groups and the Kruskal–Wallis H test for comparison between the four groups. In all cases, it was assumed that our data was not normally distributed because there were only four cases per group.

### 4.5. Protein Extraction and Western Blot (WB)

Total protein was extracted by lysing the cells with 1% sodium dodecyl sulfate, incubating at 95 °C for 15 min, and vortexing for 15 min. Protein samples were quantified using a Pierce BCA Protein Assay Kit (23225, Thermo Scientific, Waltham, MA, USA) and separated in a 10% acrylamide stain-free gel (1610183, Bio-Rad, Hercules, CA, USA) using BlueStar Pre-stained Protein Marker Plus (MWPO4, Nippon Genetics, Düren, Germany) for 30 min at 80 V and 1 h at 160 V. Stain-free gels were exposed to UV light before protein transfer to activate the trihalo compound that reacted with tryptophan residues, allowing rapid fluorescent detection of total protein. Proteins were transferred from gels to polyvinylidene fluoride membranes (10600023, GE Healthcare Life Sciences, Boston, MA, USA) for 2 h at 80 V and 4 °C. The membrane was exposed to UV light to obtain a protein charge measurement to normalize the results. Membranes were blocked with phosphate-buffered saline (PBS) with 0.1% Tween and 5% non-fat milk for 1 h at room temperature and then incubated with the primary antibody anti-desmoplakin 1/2 (2722-5204, Bio-Rad, Hercules, CA, USA) at 1:500, anti-plakoglobin (13-8500, Invitrogen, Waltham, MA, USA) at 1:1000, anti-plakophilin-2 (ab189323, Abcam, Cambridge, UK) at 1:250, anti-desmoglein2 (ab150372, Abcam, Cambridge, UK) at 1:3000, anti-desmocolin-2 (AF7490, Bio-Techne RD systems, Minneapolis, MN, USA) at 1:200, anti-ryanodine receptor-2 (NBP1-19484, Novus Biologicals, Centennial, CO, USA) at 1:1000, anti-connexin-43 (C6219, Sigma, St Louis, MO, USA) at 1:4000, anti-ankiryn-2 (821501, BioLegend, San Diego, CA, USA) at 1:200, and anti-voltage-gated sodium channel 1.5 (23016-1-AP, Proteintech, Minneapolis, MN, USA) at 1:5000 overnight at 4 °C. After several PBS washes, the membranes were incubated with peroxidase-conjugated anti-rabbit antibody (111-035-003, Jackson ImmunoResearch, West Grove, PA, USA) or anti-mouse antibody (115-035-003, Jackson ImmunoResearch, West Grove, PA, USA) and anti-goat antibody (705-035-003, Jackson ImmunoResearch, West Grove, PA, USA) at a 1:10,000 dilution for 1 h at room temperature. A chemiluminescent signal was obtained with a substrate (1705061, Bio-Rad, Hercules, CA, USA) and detected using the ChemiDoc MP imaging system. Expression levels were quantified by Image Lab software using the total protein of the stain-free blots to normalize the bands [55]. The obtained results were analyzed statistically using SPSS by performing Mann–Whitney U tests only on the blots in which bands were detected in all samples. In all cases, it was assumed that our data was not a normal distribution because there were only four cases per group. The WB bar plots that were generated represented the protein levels of triplicate sample quantification, including data from the blots shown in Figure 2 and Figure 3, and Figure 5 and also for replicates shown in the Appendix A.

### 4.6. Calcium Imaging

HL1 cells (384 cells/mm2) were seeded on register chambers (BT-CSR1, Cell Microcontrols, Norfolk, VA, USA) 24–48 h prior to the experiments. Cells were loaded with 5 µM of Fluo4-AM (F14201, Invitrogen, Waltham, MA, USA) in the presence of 0.02% Pluronic F-127 (P2443 Sigma-Aldrich, Saint Louis, MO, USA) in NK physiological solution (140 mM NaCl, 3 mM KCl, 10 mM HEPES, 1.2 mM MgCl2, 1.8 mM CaCl2, and 10 mM glucose) for 30 min at room temperature. For the calcium transient measurements, all the solutions were maintained at 35–37 °C with a temperature controller (TC2bip, Cell Microcontrols, Norfolk, VA, USA). Calcium transients were induced by a 1-s pulse of 10 mM caffeine applied with a local perfusion pipette (MPRE8, Cell Microcontrols, Norfolk, VA, USA). Calcium transients were recorded at 32 fps using an inverted microscope (Nikon Eclipse Ti, Japan) equipped with an electron-multiplying CCD camera (Hamamatsu C900-13, Hamamatsu, Japan) and a fast illumination system (Lambda DG4, Sutter Instrument, Novato, CA, USA). The acquisition was controlled from a personal computer with MetaFluor Imaging System software (Molecular Devices, Sunnyvale, CA, USA). Peak parameters were measured with Clampfit 10.2 (Molecular Devices, Sunnyvale, CA, USA), and a hierarchical statistical analysis was performed with a Ratonly RStudio script following the criteria described previously [56].

## 5. Conclusions

This study presented the first systematic in vitro study using CRISPR-edited HL1 cells to define common molecular and functional effects of PKP2, DSG2, DSC2, and DSP loss using the same experimental approach.

Our study revealed that DSC2 downregulation was the only common alteration in desmosomal expression genes shared by the absence of PKP2, DSG2, DSC2, or DSP. Interestingly, PKP2 loss triggered major alterations in desmosomal and calcium-handling gene expression. Regarding electrical conduction alterations, PKP2, DSG2, and DSC2-KO caused the downregulation of Nav1.5, but this was not shared by DSP-KO. Moreover, upregulation of TFGB1 and PPARγ was also found to be a common molecular feature for PKP2, DSG2, and DSC2 loss, although no data was available for the loss of DSP.

Finally, our results showed massive dysregulation in calcium handling genes shared by PKP2, DSC2, or DSG2 loss associated with a slower calcium re-uptake that might be related to the ANK2, CASQ2, ATP2A2, and RYR2 decrease or PLN increase. In contrast, DSP-KO clones produced a shorter amplitude of the calcium peak, which might be associated with the downregulation of PLN, which was only shown in DSP-KO clones. More studies would be needed to corroborate these associations. Moreover, the present study demonstrated that N-DSP proteins are fully functional in terms of calcium cycling.

Taking all the above into account, it would be interesting for future studies to investigate deeply the calcium handling alterations caused by the absence of desmosomal genes. After seeing the presented results in that direction, describing the association between ACM clinical features, such as arrhythmia or heart failure, and the causal gene explained by alterations of calcium handling would clarify the molecular pathophysiology of the disease. Moreover, we have proposed several molecular alterations in calcium handling that could explain the functional dysregulation. Still, more studies are needed to confirm those associations and the cause-consequence relation. Targeting the genes whose altered expression levels are the cause of presenting a shorter or delayed amplitude of the calcium peak could be useful to start searching for a treatment to compensate for the calcium handling alterations and maybe to avoid an important part of arrhythmogenic features.

## Figures and Tables

**Figure 1 ijms-24-02109-f001:**
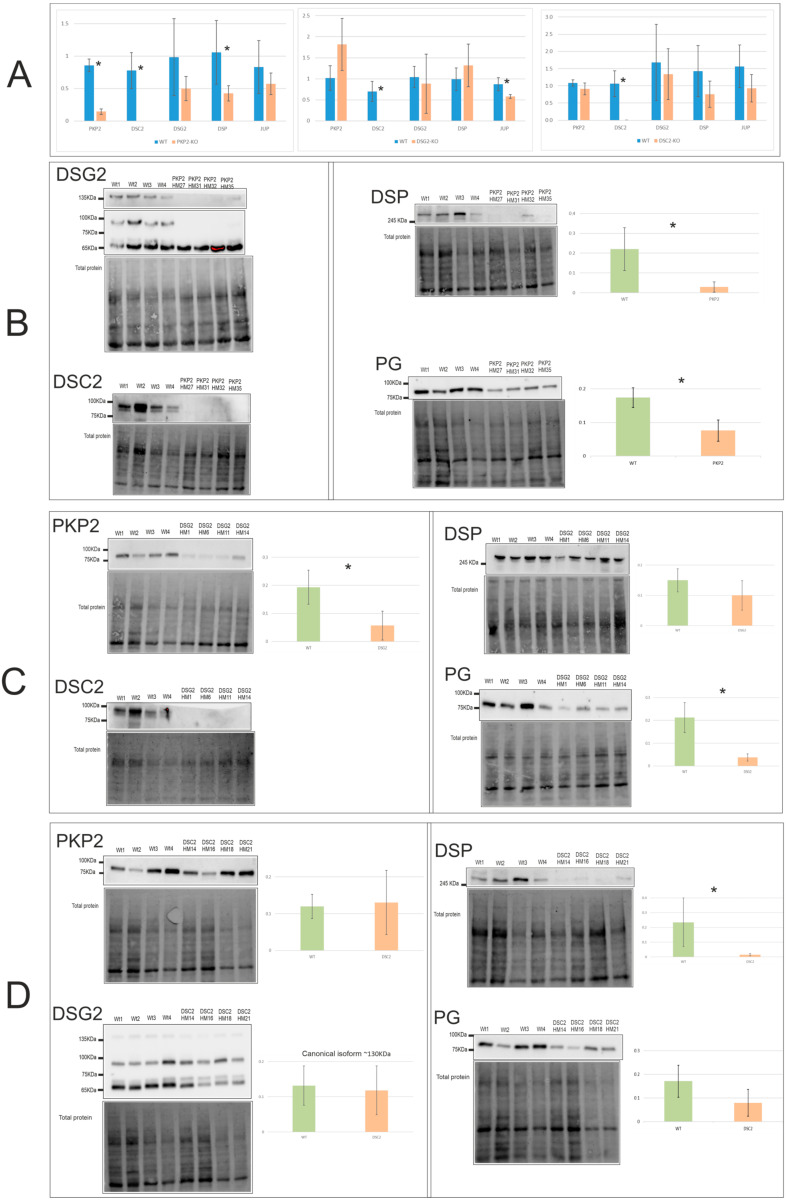
Desmosomal expression levels of mRNA (**A**) and protein of the edited clones *PKP2*-KO (**B**), *DSG2*-KO (**C**), and *DSC2*-KO (**D**). * Indicates the gene expression levels that were significantly different between WT and edited clones; *p*-value < 0.05, see Appendix A. Number of experiments: RT-PCR = 1; WB = 3.

**Figure 2 ijms-24-02109-f002:**
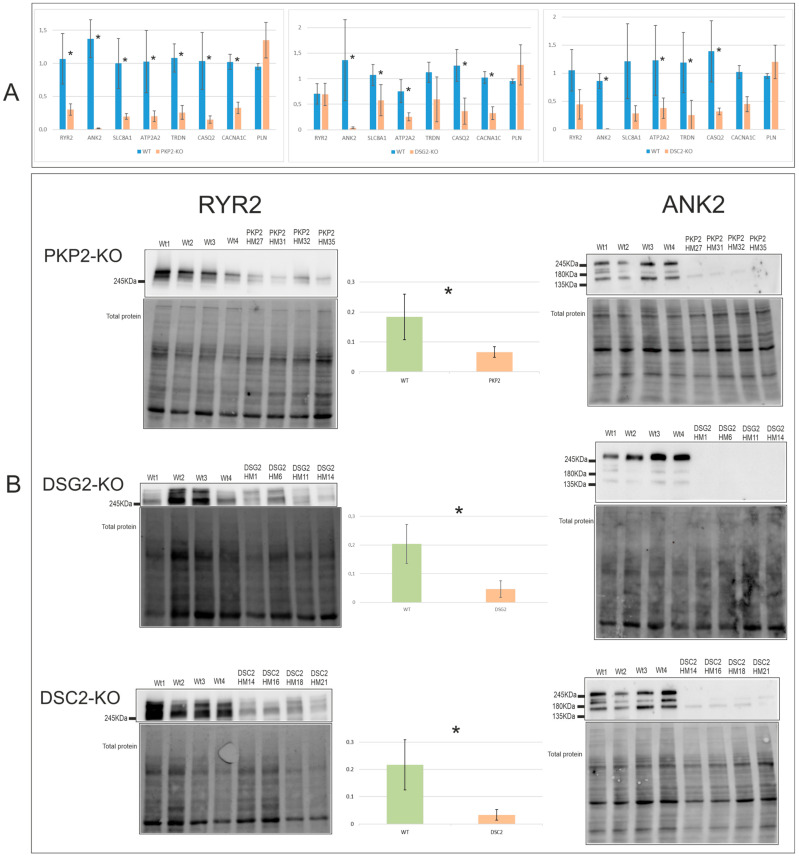
Calcium cycle gene expression at the mRNA (**A**) and protein (**B**) levels. * Indicates the gene expression levels that were significantly different between WT and edited clones; *p*-value <0.05, see Appendix A. Number of experiments: RT-PCR = 1; WB = 3.

**Figure 3 ijms-24-02109-f003:**
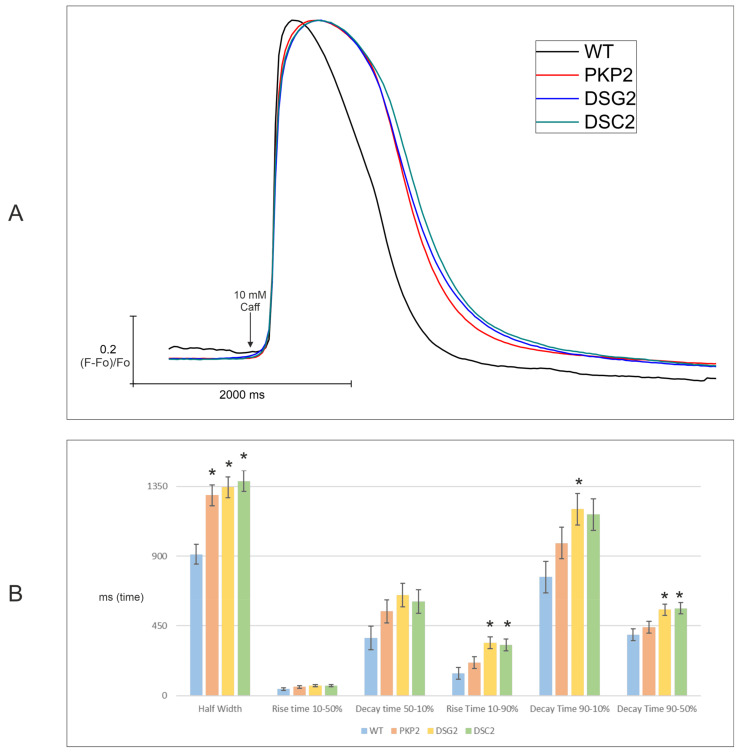
Calcium imaging peaks (**A**) and studied parameters (**B**) of the three KO groups: *PKP2*-KO, *DSG2*-KO, and *DSC2*-KO. The black arrow in section A indicates the pulse of caffeine. * Indicates parameters that were significantly different compared with WT clones; *p*-value < 0.05, see Appendix A.

**Figure 4 ijms-24-02109-f004:**
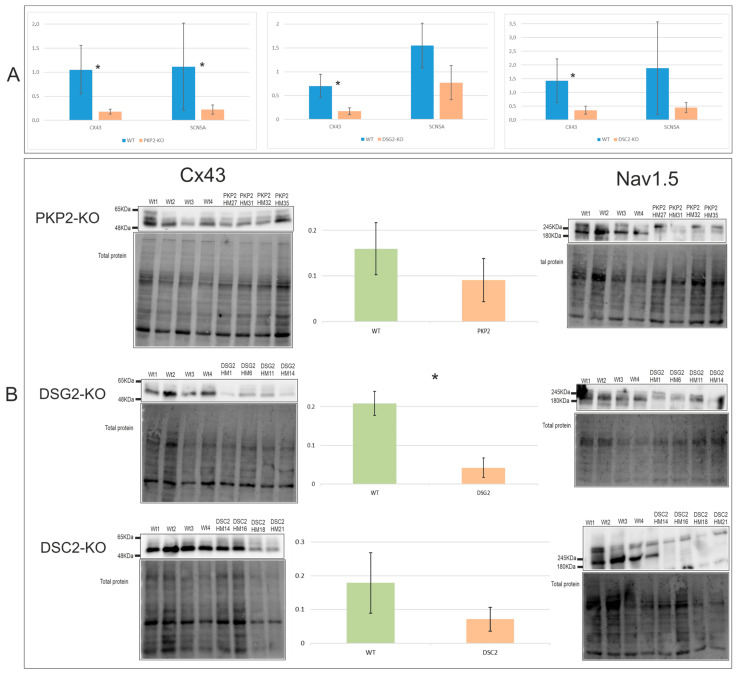
Cx43 and SCN5A expression at the mRNA (**A**) and protein (**B**) levels of the three KO groups *PKP2*-KO, *DSG2*-KO, and *DSC2*-KO. * Indicates parameters that were significantly different compared with WT clones; *p*-value < 0.05, see Appendix A. Number of experiments: RT-PCR = 1; WB = 3.

**Figure 5 ijms-24-02109-f005:**
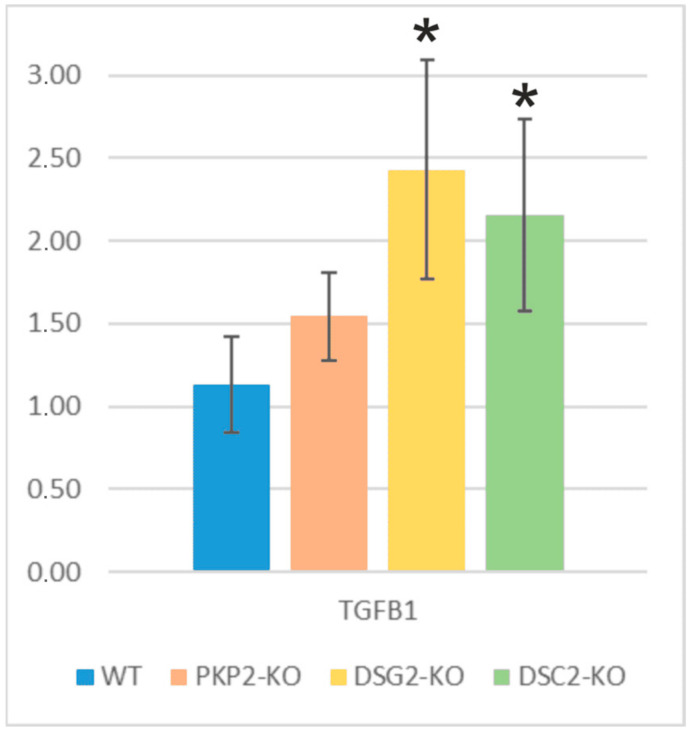
mRNA expression levels of several genes involved in ACM molecular pathways in the edited clones. * Indicates gene expression levels that were significantly different compared with WT clones; *p*-value < 0.05, see Appendix A. Number of experiments: RT-PCR = 1.

**Figure 6 ijms-24-02109-f006:**
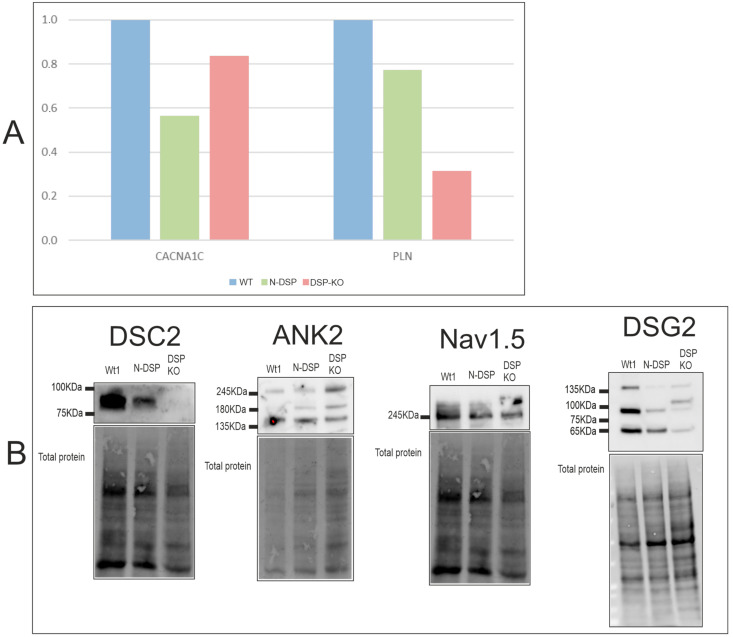
mRNA expression levels of *CACNA1C* and *PLN* (**A**) and protein levels of DSC2, ANK2, Nav1.5, and DSG2 (**B**) in DSP clones. Number of experiments: RT-PCR = 1; WB = 1.

**Figure 7 ijms-24-02109-f007:**
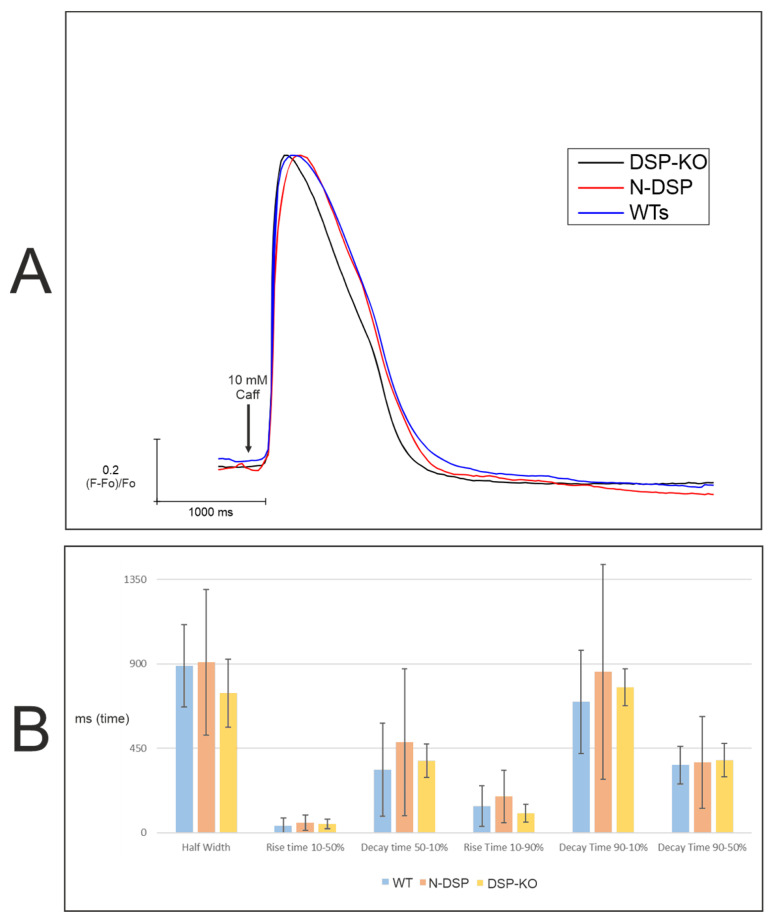
Calcium imaging peaks (**A**) and studied parameters (**B**) of the *DSP* clones. The black arrow in section A indicates the pulse of caffeine.

## Data Availability

Not applicable.

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
