# Peer review of "Alterations in Calcium Handling Are a Common Feature in an Arrhythmogenic Cardiomyopathy Cell Model Triggered by Desmosome Genes Loss"

_ijms, 2023, doi:10.3390/ijms24032109_

Round 1

Reviewer 1 Report

1.      The authors includes several PKP2-KO, DSG2-KO and DSC2-KO HL-1 clones and indicated that HL-1 was the first ACM model. It is still important to note that HL-1 is a mouse cardiomyocyte cell line. Several studies have been published using human iPSC-derived cardiomyocytes carrying these ACM-related genetic mutations. The authors should at least address the representation of their key findings in human cardiomyocytes carrying ACM-related mutations.

2.      One main challenge of in vitro studies using cardiomyocytes, regardless of murine or human cell lines, is the maturation status of those cells. Many efforts have been put to drive the maturity of cultured cardiomyocytes toward adult-like status. However, I missed this part in the cell culture section. Besides, images and/or videos of each cell line to show their status before the calcium imaging or RNA extraction are also missing.

3.      The authors examined mRNA and protein levels of five desmosomal genes in all clones, including DSP and JUP. However, I missed the rationale for selecting them specifically, and it should be added before showing the data.

4.      The authors demonstrated the altered calcium cycle by evaluating mRNA and protein levels of calcium handling-related genes and proteins in all clones, such as ATP2A2 and PLN. ATP2A2 encodes the sarcoplasmic/endoplasmic reticulum calcium-pumping ATPase (SERCA2), which plays a key role in calcium handling. ATP2A2/SERCA2 is generally suppressed in heart failure, including ACM, DCM, HCM, etc, which is in line with the findings shown by the authors. PLN binds to ATP2A2/SERCA2 to facilitate the calcium uptake and release, which was not significantly altered in those clones. Could the authors further integrate these data and elaborate on it?

5.      The authors also addressed the role of PLN in the discussion part. Interestingly, mutated PLN is also tightly associated with ACM, such as PLN R14del related ACM. Can the authors also discuss mutated PKP2, DSG2, and DSC2 related ACM with ACM carrying other common mutated genes?

Author Response

  1. The authors includes several PKP2-KO, DSG2-KO and DSC2-KO HL-1 clones and indicated that HL-1 was the first ACM model. It is still important to note that HL-1 is a mouse cardiomyocyte cell line. Several studies have been published using human iPSC-derived cardiomyocytes carrying these ACM-related genetic mutations. The authors should at least address the representation of their key findings in human cardiomyocytes carrying ACM-related mutations.

We mentioned that this was the HL-1 was the first systematic ACM model in meaning that this our study included different KO desmosomal genes in the same cellular model which it is the first time. However, we agree with the reviewer that iPSC-derived cardiomyocytes have been used to study specific ACM-related genetic mutations and we included some citations in our publications since iPSC-derived cardiomyocytes have been used to study ACM features and are they an important human cell model to reproduce ACM phenotype in vitro. In our study, we have been guided by some of iPSC studies such as:

Cited in the manuscript as:

  1. Khudiakov, A.; Zaytseva, A.; Perepelina, K.; Smolina, N.; Pervunina, T.; Vasichkina, E.; Karpushev, A.; Tomilin, A.; Malashicheva, A.; Kostareva, A. Sodium Current Abnormalities and Deregulation of Wnt/β-Catenin Signaling in IPSC-Derived Cardiomyocytes Generated from Patient with Arrhythmogenic Cardiomyopathy Harboring Compound Genetic Variants in Plakophilin 2 Gene. Biochimica et Biophysica Acta (BBA) - Molecular Basis of Disease 2020, 1866, 165915, doi:10.1016/j.bbadis.2020.165915.

  1. Hawthorne, R.N.; Blazeski, A.; Lowenthal, J.; Kannan, S.; Teuben, R.; DiSilvestre, D.; Morrissette-McAlmon, J.; Saffitz, J.E.; Boheler, K.R.; James, C.A.; et al. Altered Electrical, Biomolecular, and Immunologic Phenotypes in a Novel Patient-Derived Stem Cell Model of Desmoglein-2 Mutant ARVC. JCM 2021, 10, 3061, doi:10.3390/jcm10143061.

  1. One main challenge of in vitro studies using cardiomyocytes, regardless of murine or human cell lines, is the maturation status of those cells. Many efforts have been put to drive the maturity of cultured cardiomyocytes toward adult-like status. However, I missed this part in the cell culture section. Besides, images and/or videos of each cell line to show their status before the calcium imaging or RNA extraction are also missing.

HL1 cell line maintains the differentiated cardiac properties at morphological, biochemical and electrophysiological level. There is no protocol used to promote maturation on HL1 because they are already adult cardiomyocytes [1]. We add a representative image of our Hl1 cell culture, as an example. There were no morphological differences among the edited lines. Several studies have been published in the field using same culture protocol for HL1 cells [2-5]:

  1. Claycomb, W.C.; Lanson, N.A.; Stallworth, B.S.; Egeland, D.B.; Delcarpio, J.B.; Bahinski, A.; Izzo, N.J. HL-1 Cells: A Cardiac Muscle Cell Line That Contracts and Retains Phenotypic Characteristics of the Adult Cardiomyocyte. Proc. Natl. Acad. Sci. U. S. A. 1998, 95, 2979–2984, doi:10.1073/PNAS.95.6.2979.
  2. Gurha, P.; Chen, X.; Lombardi, R.; Willerson, J.T.; Marian, A.J. Knockdown of Plakophilin 2 Downregulates MiR-184 Through CpG Hypermethylation and Suppression of the E2F1 Pathway and Leads to Enhanced Adipogenesis In Vitro. Circ. Res. 2016, 119, 731–750, doi:10.1161/CIRCRESAHA.116.308422.
  3. Shoykhet, M.; Trenz, S.; Kempf, E.; Williams, T.; Gerull, B.; Schinner, C.; Yeruva, S.; Waschke, J. Cardiomyocyte Adhesion and Hyperadhesion Differentially Require ERK1/2 and Plakoglobin. JCI insight 2020, 5, doi:10.1172/JCI.INSIGHT.140066.
  4. Agullo-Pascual, E.; Cerrone, M.; Delmar, M. Arrhythmogenic Cardiomyopathy and Brugada Syndrome: Diseases of the Connexome. FEBS Lett 2014, 588, 1322–1330, doi:10.1016/j.febslet.2014.02.008.
  5. Cerrone, M.; van Opbergen, C.J.M.; Malkani, K.; Irrera, N.; Zhang, M.; Van Veen, T.A.B.; Cronstein, B.; Delmar, M. Blockade of the Adenosine 2A Receptor Mitigates the Cardiomyopathy Induced by Loss of Plakophilin-2 Expression. Front. Physiol. 2018, 9, doi:10.3389/FPHYS.2018.01750.
  6. The authors examined mRNA and protein levels of five desmosomal genes in all clones, including DSP and JUP. However, I missed the rationale for selecting them specifically, and it should be added before showing the data.

We examined the expression levels of all desmosomal genes in all clones because it has been reported altered expression in some of them among several cellular models carrying different mutations. We aimed to study the impact on the expression of every desmosome protein triggered by each desmosomal gene loss. We wanted to detect alterations in all clones to potentially identify gene-specific deficits in desmosomal protein expression.

  1. The authors demonstrated the altered calcium cycle by evaluating mRNA and protein levels of calcium handling-related genes and proteins in all clones, such as ATP2A2 and PLN. ATP2A2 encodes the sarcoplasmic/endoplasmic reticulum calcium-pumping ATPase (SERCA2), which plays a key role in calcium handling. ATP2A2/SERCA2 is generally suppressed in heart failure, including ACM, DCM, HCM, etc, which is in line with the findings shown by the authors. PLN binds to ATP2A2/SERCA2 to facilitate the calcium uptake and release, which was not significantly altered in those clones. Could the authors further integrate these data and elaborate on it?

We added the following paragraph (line 375):

“Regarding the relation between ACM clinical features and these molecular and functional detected alterations in calcium handling, it is interesting to add some related data. On the one hand, it has been observed an association between decreased SERCA2 and heart failure [50]. In that sense, it has been developed a gene therapy for heart failure based on the overexpression of SERCA2. There is data that confirm its effectiveness [51,52], but, for the moment, it has not been validated in a large-scale clinical trial [53,54]. In the present study, PKP2, DSG2 and DSC2-KOs experiment a decrease on ATPase expression (codifies for SERCA) indicating that the absence of those genes may be linked to heart failure, a clinical feature present in ACM phenotype.

On the other hand, the molecular and functional detected alterations may also be related to dysregulation on the Ca2+ content, that is associated with different clinical alterations [43]. A high Ca2+ content in the SR could contribute to propagate wave of Ca2+ induced - Ca2+ release increasing the propensity to arrhythmias while a low Ca2+ content is associated with heart failure [43]. Our experimental approach was not able to determine the Ca2+ content of each cell line, but for future experiments it would be interesting to measure it to see if there is any association between arrhythmias or heart failure with the loss of a determinate desmosomal protein via calcium handling.”

  1. The authors also addressed the role of PLN in the discussion part. Interestingly, mutated PLN is also tightly associated with ACM, such as PLN R14del related ACM. Can the authors also discuss mutated PKP2, DSG2, and DSC2 related ACM with ACM carrying other common mutated genes?

We added the following paragraph (line 365):

“Moreover, it is important to take into account that PLN is an ACM-associated gene classified as definite by ClinGen [45]. Specifically, there are several studies of the variant PLN R14del which is prevalent in the Netherlands and it presents a stronger affinity for SERCA2, suggesting that the inhibition of SERCA2 is higher on those patients and so it is the risk for malignant ventricular arrhythmias [46–48]. Regarding its role in calcium handling, it has been shown that causes a slower SR Ca+2 reuptake [49]. Taking this into account, the calcium handling dysregulation underlying ACM caused by PLN R14del or by the absence of PKP2, DSG2 or DSC2 might be similar. More studies are needed on that direction to elucidate if those ACM-causal genes share molecular and functional alterations on calcium handling.”

Reviewer 2 Report

Etiopathogenesis of arrhythmogenic cardiomyopathy is not comprehensively elucidated despite causative implication of desmosomal gene defects along with associated interacting proteins.  It is highly probable that not only gene-specific molecular alterations but also epigenetic factors and overall “connexome”, including Cx and Nav channels (see f.e. Agullo-Pascual 2014, Ben-Haim 2021) implicated in remodelling of intercalated disc (see f.e. Vite 2014, Tribulova 2015) as well as yet unravelled factors may be involved in this pro-arrhythmic cardiomyopathy. Likely including the Cx43 hemichannels (Andelova 2021).

Thus, it is appreciated that current article using HL1 desmosomal gene knockout cells identified molecular and functional alterations in calcium handling (slower calcium reuptake). These disorders may be explained by altered mRNA expression levels of calsequestrin-2, ATPase sarcoplasmic/endoplasmic reticulum Ca2+ transporter-2, ryanodine receptor-2 and phospholamban. It is step forward in understanding mechanisms of inherited arrhythmogenic cardiomyopathy. Nevertheless, there are some proposals/comments to be addressed prior publishing.

Abstract: Conclusive sentence pointing out impact or association of Ca2+ handling alterations with desmosomal defects on development of arrhythmias is missing in the abstract.

Introduction: It would be worthwhile to include some references (noted in first paragraph) pointing out the role intercalated disc “connexome”, i.e. mutual interaction of adhesive junctions and gap junctions (Cx43 channels in arrhythmogenesis as well possible implication of Cx43 hemichannels.

Results:  WB shows two proteins in Figure 2B and 4B but  data from which protein are plotted on graphs??? One is missing???

PPARg should be displaced by PPARγ

Discussion: How we should understand that down regulation od Ca2+ handling, including Ryr2 is pro-arrhythmic? Due to delay in Ca reuptake by sarcoplasmic reticulum? It is well known that Ca2+ leaks via Ryr2 is pro-arrhythmic or Ca2+ overload due to defects in Ca2+ uptake by sarcoplasmic reticulum found in many cardiac conditions. Is it some common feature?

Findings point out the implication of “connexome” and its link with calcium handling, what you think about?

It would be appreciated to include Conclusions and perspectives: 

Maybe further in vivo studies to unravel association of desmosomal defects with Ca handling disorders. And what about the treatment to hamper adverse signalling of desmosomal defects?

Author Response

Etiopathogenesis of arrhythmogenic cardiomyopathy is not comprehensively elucidated despite causative implication of desmosomal gene defects along with associated interacting proteins.  It is highly probable that not only gene-specific molecular alterations but also epigenetic factors and overall “connexome”, including Cx and Nav channels (see f.e. Agullo-Pascual 2014, Ben-Haim 2021) implicated in remodelling of intercalated disc (see f.e. Vite 2014, Tribulova 2015) as well as yet unravelled factors may be involved in this pro-arrhythmic cardiomyopathy. Likely including the Cx43 hemichannels (Andelova 2021).

Thus, it is appreciated that current article using HL1 desmosomal gene knockout cells identified molecular and functional alterations in calcium handling (slower calcium reuptake). These disorders may be explained by altered mRNA expression levels of calsequestrin-2, ATPase sarcoplasmic/endoplasmic reticulum Ca2+ transporter-2, ryanodine receptor-2 and phospholamban. It is step forward in understanding mechanisms of inherited arrhythmogenic cardiomyopathy. Nevertheless, there are some proposals/comments to be addressed prior publishing.

Abstract: Conclusive sentence pointing out impact or association of Ca2+ handling alterations with desmosomal defects on development of arrhythmias is missing in the abstract.

We added the following sentence in the abstract (line 27)

“In conclusion, the loss of desmosomal genes provokes alterations in calcium handling poten-tially contributing to the development of the arrhythmogenic events in ACM.

Introduction: It would be worthwhile to include some references (noted in first paragraph) pointing out the role intercalated disc “connexome”, i.e. mutual interaction of adhesive junctions and gap junctions (Cx43 channels in arrhythmogenesis as well possible implication of Cx43 hemichannels.

We added the following paragraph (line 50):

“In that sense, it has been described that electrical coupling and intercellular adhesion in the heart are connected by a functional unit called connexome, it is constituted by voltage-gated sodium channel (Nav1.5), Connexin-43 (Cx43) and desmosomes [8,9]. In fact, it is known that there is an overlapping between Brugada Syndrome and ACM due to these interactions found in the connexome [10,11].”

Results:  WB shows two proteins in Figure 2B and 4B but data from which protein are plotted on graphs??? One is missing???

Only those proteins with detectable levels of protein in the WBs are plotted in graphs. In figure 2B, RYR2 protein levels are plotted, but not ANK2 levels because this protein showed undetectable levels on the KO clones. These two protein results are separated by a line to divide the expression levels of RYR2 and ANK2. Representation of Figure 4B is the same case as Figure 2B, CX43 levels are plotted, but not the Nav1.5 because it was not detected on the KO clones.  We also added this information in the method sections to clarify this point (line 484)

PPARg should be displaced by PPARγ

 We change it in the whole manuscript and supplemental material.

Discussion: How we should understand that down regulation of Ca2+ handling, including Ryr2 is pro-arrhythmic? Due to delay in Ca reuptake by sarcoplasmic reticulum? It is well known that Ca2+ leaks via Ryr2 is pro-arrhythmic or Ca2+ overload due to defects in Ca2+ uptake by sarcoplasmic reticulum found in many cardiac conditions. Is it some common feature? Findings point out the implication of “connexome” and its link with calcium handling, what you think about?

We added the following paragraph (line 375):

“Regarding the relation between ACM clinical features and these molecular and functional detected alterations in calcium handling, it is interesting to add some related data. On the one hand, it has been observed an association between decreased SERCA2 and heart failure [50]. In that sense, it has been developed a gene therapy for heart failure based on the overexpression of SERCA2. There is data that confirm its effectiveness [51,52], but, for the moment, it has not been validated in a large-scale clinical trial [53,54]. In the present study, PKP2, DSG2 and DSC2-KOs experiment a decrease on ATPase expression (codifies for SERCA) indicating that the absence of those genes may be linked to heart failure, a clinical feature present in ACM phenotype.

On the other hand, the molecular and functional detected alterations may also be related to dysregulation on the Ca2+ content, that is associated with different clinical alterations [43]. A high Ca2+ content in the SR could contribute to propagate wave of Ca2+ induced - Ca2+ release increasing the propensity to arrhythmias while a low Ca2+ content is associated with heart failure [43]. Our experimental approach was not able to determine the Ca2+ content of each cell line, but for future experiments it would be interesting to measure it to see if there is any association between arrhythmias or heart failure with the loss of a determinate desmosomal protein via calcium handling.”

It would be appreciated to include Conclusions and perspectives: 

Maybe further in vivo studies to unravel association of desmosomal defects with Ca handling disorders. And what about the treatment to hamper adverse signalling of desmosomal defects?

We added the following paragraph (line 528):

“Taking all the above into account, it would be interesting for future studies to investigate deeply the calcium handling alterations caused by the absence of desmosomal genes. After seeing the presented results on that direction, describing the association between ACM clinical features, such as arrhythmias or heart failure, and the causal gene explained by alterations of calcium handling would clarify the molecular pathophysiology of the disease. Moreover, here we have proposed several molecular alterations on calcium handling that could explain the functional dysregulation, but more studies are needed to confirm those associations and the cause-consequence relation. Targeting the genes whose altered expression levels are the cause of presenting a shorter or delayed amplitude of the calcium peak could be useful to start searching for a treatment to compensate the calcium handling alterations and maybe to avoid an important part of arrhythmogenic features.”
